# PROVABLE GUARANTEES FROM PRACTICAL REGULARIZATION FOR ALIGNMENT WITH HUMAN PREFERENCES

## ABSTRACT

Reinforcement learning with human preference feedback is the gold-standard approach for making current AI systems helpful, safe, and aligned with human values. Recent research has demonstrated that there is a tight connection between the objective functions used for alignment with human preferences, and voting rules from social choice theory that aggregate diverse preferences. This connection provides a principled way to study the advantages and disadvantages of a given alignment objective by analyzing the social-choice theoretic properties of the corresponding voting rule. Prior work in this direction has focused on variants of standard alignment objective functions, and connected them with well-known social choice rules such as the Borda count and von Neumann winner rules. However, practical alignment algorithms typically perform regularization to a reference policy in order to maintain the capabilities from pre-training. Such regularization could potentially distort the objective and hence change the social-choice theoretic properties of the corresponding voting rule. To address this question, we study the effect of regularization on the social-choice rules corresponding to standard alignment methods, and discover that in the case of the alignment objective corresponding to the von Neumann winner, regularization strictly improves the social-choice theoretic properties of the rule. At the same time, we prove that the standard RLHF objective, which corresponds to the Borda count rule, offers no such improvement and indeed has clear social-choice theoretic drawbacks compared to the von Neumann winner. Taken together, our results provide principled justification from social choice theory to use the von Neumann winner objective for practical alignment with human preferences.

## 1 INTRODUCTION

Recent work on alignment of AI systems via human preference rankings has drawn strong connections to social choice theory. The analogy is as follows: one can view the possible outputs of an AI model as a set of *candidates*, human raters as a set of *voters*, and the algorithm used for alignment as defining a rule to aggregate the individual voter preference rankings over the candidates. Thus the problem of designing algorithms for aligning AI systems becomes a problem of designing social choice rules. The advantage of this social-choice viewpoint is that it explicitly considers cases where voters express legitimate differences of opinion in their preferences, going beyond many standard models in machine learning which consider each human rating as a sample from a single, coherent ground-truth preference ranking (Huang et al., 2024). By taking pluralistic value alignment into account, we can foster a more democratic paradigm for human-AI interactions.

Quite recently, there has been substantial progress in analyzing alignment methods with the lens of social choice theory. Standard approaches to reinforcement learning from human feedback (RLHF) (Christiano et al., 2017; Bai et al., 2022) based on the Bradley-Terry model (Bradley & Terry, 1952) have been shown to implicitly apply the Borda count social choice rule when aggregating human preference rankings (Siththaranjan et al., 2024). That is, the optimal model output under the standard RLHF objective is equal to the highest ranked output according to the Borda count rule when considering raters as voters. Various alternative methods have also been proposed, including a recent line of work utilizing the solution concept known as the von Neumann winner (Dudík et al., 2015; Wang et al., 2023; Munos et al., 2024; Swamy et al., 2024), which optimizes an objective equivalent to the randomized social choice rule known as *maximal lotteries*.

In general, the theory of social choice takes an axiomatic approach to the selection of social choice rules, with a focus on trade-offs between desirable axiomatic properties as exemplified by Arrow's theorem (Arrow, 1951). The key axioms studied in social choice theory include many that are naturally desirable in the context of AI alignment. *Independence of irrelevant alternatives* ensures that the final policy is invariant to changing preferences over irrelevant potential outputs that the model is very unlikely to produce. *Population consistency* ensures that merging two sets of ratings, each of which individually yield similar policies, will not cause a sudden change in the behavior of the model. The requirement that a rule is *strategy-proof* ensures that no rater can influence the final policy for their own gain by strategically misreporting their preferences.

While it is tempting to directly apply results from social choice in order to choose between alignment methods, such an approach leaves out key components of standard algorithmic pipelines. In particular, algorithms for alignment from preference rankings typically include a KL-divergence regularizer, to ensure that the aligned policy does not deviate too far from some fixed reference policy. This regularization is necessary in order to prevent the policy from straying so far that it forgets how to write grammatical sentences, but also has an impact on how human preferences are aggregated.

In this paper, we investigate AI alignment within the context of social choice theory and ask the following question:

> *What is the impact of explicitly preserving the initial pre-trained policy on alignment from the perspective of axiomatic social choice?*

## 1.1 Our Approach.

We model the problem of aligning artificial intelligence to human preferences as a problem of social choice theory. The alternatives for the output of the model play the role of candidates and individual human raters play the role of voters. The overall goal of an alignment algorithm is to output a distribution over candidates that aggregates human preferences in a principled way.

**AI Alignment as probabilistic social choice.** Our model of alignment directly corresponds to the setting of *probabilistic social choice*, where rather than producing an aggregate ranking over candidates, the social choice rule produces an aggregate ranking over distributions over candidates. The probabilistic setting is quite natural in the case of aligning LLMs, where the model already produces a distribution over possible responses, rather than one deterministic outcome.

The probabilistic social choice model of alignment is more general than that proposed in several recent papers on social choice for alignment. A line of prior work that considers axiomatic social choice in alignment assumes that the alignment algorithm initially learns a *deterministic* reward function that aggregates individual preferences (Siththaranjan et al., 2024; Ge et al., 2024). After the aggregate reward function is learned, it is assumed that standard reinforcement learning methods are used to train a policy distribution that maximizes this learned reward. Due to the assumption that a deterministic reward function is learned as an intermediate step, this line of prior work inherently restricts itself to the setting of *deterministic social choice*. While this restriction is natural, in the sense that it corresponds to the standard setup of the original RLHF algorithm as it is applied to LLM alignment tasks, it brings with it all the limitations known to exist for deterministic social choice.

There are many cases where known impossibility results for deterministic social choice can be circumvented in the probabilistic social choice setting. Most notably, Arrow's theorem (Arrow, 1951) proves that any deterministic social choice rule that satisfies pareto efficiency and independence from irrelevant alternatives is a dictatorship. On the other hand, Brandl & Brandt (2020) recently proved that Arrow's impossibility can be avoided in the setting of probabilistic social choice. They showed that, not only is it possible to satisfy Arrow's axioms without resorting to dictatorship, but the *maximal lotteries* social choice rule (Kreweras, 1965; Fishburn, 1984) is the unique rule satisfying the axioms while utilizing anonymous votes (which precludes dictatorship).

Interestingly, the alignment objective known as the von Neumann winner (Dudík et al., 2015; Wang et al., 2023; Munos et al., 2024) corresponds precisely to the maximal lotteries rule. There are by now several different algorithms proposed to optimize a version of the von Neumann winner, and hence many different practical approaches for training LLMs that enjoy the axiomatic advantages of

the maximal lotteries rule (Wang et al., 2023; Munos et al., 2024; Rosset et al., 2024; Swamy et al., 2024).

**The role of regularization.** Commonly, alignment algorithms employ a KL-divergence regularizer during training to prevent the learned policy from deviating too far from a given reference policy. The reference policy is typically taken to be the initial pre-trained and instruction-tuned LLM, that has not yet undergone alignment via RLHF or other methods. While this regularization is necessary to avoid losing capabilities learned from pre-training, it clearly affects the final distribution on alternatives output by the alignment algorithm, and hence the axiomatic guarantees of the social choice rule. A priori, it seems that adding regularization to a probabilistic social choice rule could have a negative impact on the set of social choice axioms that it satisfies. However, we will demonstrate that the opposite is true in the case of the von Neumann winner rule.

### 1.2 OUR RESULTS

We prove that the KL-regularized von Neumann winner is approximately strategy-proof i.e. an individual human rater cannot change the outcome distribution by more than $O(1/n)$ in total-variation distance, where $n$ is the number of raters. This stands in contrast to the unregularized setting, where we prove that a single rater can change the outcome by a constant (independent of $n$) via strategic voting.

We further prove that the KL-regularized von Neumann winner still satisfies key properties of the unregularized rule including independence of irrelevant alternatives, population consistence, and (a regularized version of) Pareto efficiency.

Finally, we prove that KL-regularization does not resolve the issues of the Borda count rule, which remains not independent of irrelevant alternatives. The Borda count rule is especially significant in the context of alignment algorithms, as it is implicitly implemented by standard RLHF based on the Bradley-Terry preference model. Taken together, these results suggest a strong axiomatic advantage to using the KL-regularized von Neumann winner for AI alignment from human preferences.

Our results provide principled analysis on which practical alignment methods inherently satisfy axioms of social choice theory for free, with the ultimate goal of achieving fair, pluralistic AI alignment.

## 2 RELATED WORK

Along with the notable practical success of RLHF for aligning LLMs to human preferences, there has been a growing body of work studying how to apply social choice theory to RLHF algorithms, in order to take into account general human preferences that may contain legitimate disagreements on values. The work of Siththaranjan et al. (2024) pointed out a crucial connection between social choice and AI alignment: the standard RLHF objective based on the Bradley-Terry model implicitly aggregates the preferences of different raters via the Borda count social choice rule. Concurrently, the position papers of Conitzer et al. (2024); Mishra (2023) argued for the necessity of using social choice theory to inform AI alignment from human preferences. More recently, Ge et al. (2024) proposed a theoretical model for studying the social choice theoretic axioms satisfied by alignment algorithms. This paper focused on axiomatic social choice theory, in the setting of linear reward functions, where both the aggregated reward and the individual rater preferences are restricted to be linear in some feature space. While the focus on evaluating alignment rules via social choice axioms is similar to our paper, this work differs by working with deterministic, linear preferences, whereas we consider probabilistic social choice with unrestricted preference functions.

There has also been a parallel line of work that attempts to directly address the issue of general, possibly disagreeing preferences in RLHF. In particular, Wang et al. (2023); Swamy et al. (2024); Munos et al. (2024) all propose different algorithms to implement the von Neumann winner rule. Each of these papers provide explanations and examples of situations where general, incoherent preferences are better captured by the von Neumann winner than alternative rules. Wang et al. (2023) in particular mentions favorable social-choice theoretic properties of the von Neumann winner as a justification for the use of the rule. Our paper builds on this work, by demonstrating that the

regularization typically used in RL training can improve the social-choice theoretic properties of the von Neumann winner, while maintaining the advantages of the unregularized rule.

## 3 THE MODEL OF ALIGNMENT AS SOCIAL CHOICE

We assume there is a set $\mathcal{H}$ of $n$ *human raters* and a countable set $\mathcal{U}$ of *alternatives* corresponding to all potential outputs of an AI model. For an LLM, $\mathcal{U}$ is the set of all grammatically valid text of finite length. Each rater $h \in \mathcal{H}$ has an associated preference ranking $\succ_h$ over alternatives $a \in \mathcal{U}$. That is, $a \succ_h b$ means that rater $h$ prefers alternative $a$ to alternative $b$. For a subset of alternatives $\mathcal{A} \subseteq \mathcal{U}$ and a probability distribution $\pi$ over $\mathcal{A}$, we use the notation $\pi|_{\mathcal{A}}$ to denote the restriction of $\pi$ to $\mathcal{A}$ i.e. the function on $\mathcal{A}$ that assigns probability $\pi(a)$ to $a \in \mathcal{A}$.

A *probabilistic preference relation* is an anti-symmetric mapping $\mathcal{R} : \Delta\mathcal{U} \times \Delta\mathcal{U} \to \mathbb{R}$, where for any two distributions $\pi, \pi'$ over alternatives, $\mathcal{R}(\pi, \pi') \geq 0$ means that the distribution $\pi$ is preferred to the distribution $\pi'$. For a subset of alternatives $\mathcal{A} \subseteq \mathcal{U}$, we use $\mathcal{R}|_{\mathcal{A}}$ to denote the restriction of the preference relation to pairs of distributions $(\pi, \pi') \in \Delta\mathcal{A} \times \Delta\mathcal{A}$.

An *alignment rule* is a function $F$ takes as input a set of raters $\mathcal{H}$ and a subset $\mathcal{A} \subseteq \mathcal{U}$ of $m$ alternatives, and outputs a probabilistic preference relation $\mathcal{R} = F(\mathcal{A}, \mathcal{H})$ that represents an aggregation of individual rater preferences. A distribution over a subset of alternatives $\pi^* \in \Delta\mathcal{A}$ is *maximal* for a probabilistic preference relation $\mathcal{R}$ if $\mathcal{R}(\pi^*, \pi) \geq 0$ for all $\pi \in \Delta\mathcal{A}$. An alignment rule naturally induces an objective for RLHF: the goal is to train the model to output a distribution over alternatives $\pi^*$ that is *maximal* according to the probabilistic preference ranking $\mathcal{R} = F(\mathcal{A}, \mathcal{H})$.

We now turn to a set of natural probabilistic preference relations induced by the individual rater preferences. Formally, we define the *individual pairwise preference relation* to be

$$\mathcal{P}_h(\pi, \pi') = \Pr_{\substack{a \sim \pi \\ b \sim \pi'}} [a \succ b] - \Pr_{\substack{a \sim \pi \\ b \sim \pi'}} [b \succ a]$$

That is the pairwise preference relation gives the probability that the rater $h$ prefers alternatives sampled from $\pi$ to those from $\pi'$, minus the probability that the rater prefers the opposite. In particular, $\mathcal{P}_h(\pi, \pi') \geq 0$ if and only if $h$ prefers alternatives sampled from $\pi$ to those sampled from $\pi'$ on average. Observe that the pairwise preference relation is anti-symmetric i.e. $\mathcal{P}_h(\pi, \pi') = -\mathcal{P}_h(\pi', \pi)$.

Given a set of raters $\mathcal{H}$ we define the corresponding *aggregate pairwise preference relation* to be the function

$$\mathcal{P}_{\mathcal{H}}(\pi, \pi') = \mathbb{E}_{h \sim \mathcal{H}}[\mathcal{P}_h(\pi, \pi')].$$

When the relevant set of raters is clear from context we will simply write $\mathcal{P}(\pi, \pi')$. We will also use the notation $\mathcal{P}(a, b)$ for $a, b \in \mathcal{A}$ to denote the preference $\mathcal{P}(1_a, 1_b)$ where $1_a, 1_b$ are the probability distributions that puts all their mass on $a$ and $b$ respectively.

Standard reinforcement learning training utilizes KL-regularization to a reference policy. Thus, in our setting the reference policy is a distribution $\mu$ on $\mathcal{U}$, and we define the regularized preference relation

$$\widetilde{\mathcal{P}}(\pi, \pi') = \mathcal{P}(\pi, \pi') - \tau D_{\mathrm{KL}}(\pi \| \mu) + \tau D_{\mathrm{KL}}(\pi' \| \mu)$$

where $\tau > 0$ is the KL-regularization parameter.

## 4 AXIOMS FOR PROBABILISTIC SOCIAL CHOICE.

We next formally define the key social-choice axioms that we will use to evaluate alignment rules.

**Definition 4.1** (Pareto Optimality). Let $F$ be an alignment rule, and let $\mathcal{R} = F(\mathcal{A}, \mathcal{H})$. The alignment rule $F$ is *Pareto optimal* if $\mathcal{P}_h(\pi, \pi') \geq 0$ for all $h \in \mathcal{H}$ implies that $\mathcal{R}(\pi, \pi') \geq 0$.

Pareto optimality is also referred to as *unanimity* in the the social choice setting. Intuitively, Pareto optimality requires that there is no way to switch the aggregate preferences $\mathcal{R}$ for any pair $\pi, \pi'$ without making at least one rater less happy with the outcome. This intuition can be seen by taking the contrapositive of the implication in Definition 4.1.

We next turn to independence of irrelevant alternatives, an axiom which intuitively says that changes in raters' preferences about alternatives outside of some subset $\mathcal{A}$ should not affect the relative rankings of alternatives within $\mathcal{A}$.

**Definition 4.2** (Independence of irrelevant alternatives). An alignment rule $F$ satisfies independence of irrelevant alternatives if, for all subsets of alternatives $\mathcal{A} \subseteq \mathcal{B} \subseteq \mathcal{U}$, and all sets of raters $\mathcal{H}, \mathcal{H}'$ that have the same preferences over alternatives in $\mathcal{A}$ we have

$$F(\mathcal{B}, \mathcal{H})|_{\mathcal{A}} = F(\mathcal{B}, \mathcal{H}')|_{\mathcal{A}}$$

We next define an approximate notion of independence of irrelevant alternatives that can be more applicable to practical settings of training LLM policies $\pi$ from samples.

**Definition 4.3** (Approximate independence of irrelevant alternatives). Let $F$ be an alignment rule, $\mathcal{A} \subseteq \mathcal{B} \subseteq \mathcal{U}$ be subsets of alternatives, and $\mathcal{H}, \mathcal{H}'$ be any two subsets that have the same preferences over alternatives in $\mathcal{A}$. Let $\mathcal{R} = F(\mathcal{B}, \mathcal{H})|_{\mathcal{A}}$ and $\mathcal{R}' = F(\mathcal{B}, \mathcal{H}')|_{\mathcal{A}}$. The rule $F$ satisfies $\epsilon$-approximate IIA if for every $\pi, \pi'$ such that $\mathcal{R}(\pi, \pi') \geq 0$, then there exist policies $\hat{\pi}, \hat{\pi}'$ with $d_{\text{TV}}(\pi, \hat{\pi}) < \epsilon$ and $d_{\text{TV}}(\pi', \hat{\pi}') < \epsilon$ such that $\mathcal{R}'(\hat{\pi}, \hat{\pi}') \geq 0$.

Intuitively, this definition says that the relative ranking of policies over $\mathcal{A}$ produced by $F$ is not sensitive to changes in irrelevant alternatives outside of $\mathcal{A}$, so long as we are allowed to perturb the policies in question by up to $\epsilon$ in total variation distance. That is, if $\pi$ is preferred to $\pi'$ by $\mathcal{R}$, then there are two nearby policies $\hat{\pi}, \hat{\pi}'$ such that $\hat{\pi}$ is preferred to $\hat{\pi}'$ by $\mathcal{R}'$. Clearly, if an alignment rule $F$ satisfies independence of irrelevant alternatives, then it immediately satisfies $\epsilon$-approximate independence of irrelevant alternatives with $\epsilon = 0$.

We next turn to the axiom of population consistency, which requires that if two different sets of raters share a maximal policy according to the alignment rule, then that policy is still maximal when merging the two sets of raters. This axiom can be seen as a form of stability under combining sets of raters with similar aggregate preferences.

**Definition 4.4** (Population consistency). Let $\mathcal{H}$ and $\mathcal{H}'$ be two sets of raters and $\mathcal{A}$ a set of alternatives. An alignment rule $F$ satisfies *population consistency* if whenever $\pi^*$ is maximal for both $F(\mathcal{A}, \mathcal{H}), F(\mathcal{A}, \mathcal{H}')$ then $\pi^*$ is also maximal for $F(\mathcal{A}, \mathcal{H} \cup \mathcal{H}')$.

Finally, we introduce an axiom limiting the opportunity for strategic manipulation by raters. In particular, an alignment rule is approximately strategy-proof, if a subset of raters that strategically misreport their preferences cannot significantly improve their own outcomes.

**Definition 4.5** (Approximately strategy-proof). Let $k \leq n$, $\epsilon > 0$, and $\mathcal{H}$ be a set of raters. Let $\mathcal{H}_*$ be a subset of $k$ raters from $\mathcal{H}$. Let $\mathcal{H}'$ be the set where the raters $h_* \in \mathcal{H}_*$ replace their preference rankings $\prec_{h_*}$ with different preference ranking $\prec'_{h_*}$. Let $\pi$ be maximal for $F(\mathcal{A}, \mathcal{H})$ and $\pi'$ be maximal for $F(\mathcal{A}, \mathcal{H}')$. An alignment rule $F$ is $(k, \epsilon)$-approximately strategy proof if $\mathcal{P}_{\mathcal{H}^*}(\pi', \pi) \leq \epsilon$ for all choices of $\mathcal{H}, \mathcal{H}^*, \mathcal{H}'$ satisfying the above assumptions.

A rule that is $(1, 0)$-approximately strategy-proof corresponds to the classical deterministic social choice definition of strategy-proof.

## 5  THE VON NEUMANN WINNER RULE

In this section, we analyze the axiomatic properties of the regularized von Neumann winner rule. Our main results show that regularization improves the strategy-proofness of the von Neumann winner, while at the same time preserving the other key axioms that the unregularized rule satisfies. We begin with the definition of the standard von Neumann winner as follows. Given a set of alternatives $\mathcal{A}$, the von Neumann winner rule is defined by $F_{\text{vNw}}(\mathcal{A}, \mathcal{H}) = \mathcal{P}_{\mathcal{H}}|_{\mathcal{A}}$. Hence, the maximal distributions $\pi^*$ for $F_{\text{vNw}}(\mathcal{A}, \mathcal{H})$ are given by

$$\pi^* \in \arg\max_{\pi \in \Delta_{\mathcal{A}}} \min_{\pi' \in \Delta_{\mathcal{A}}} \mathcal{P}_{\mathcal{H}}(\pi, \pi'). \tag{1}$$

The interpretation of this rule is game-theoretic. For a set of alternatives $\mathcal{A}$ the pairwise preference relation $\mathcal{P}_{\mathcal{H}}|_{\mathcal{A}}$ induces a symmetric zero-sum game, where the pure strategies are alternatives $a \in \mathcal{A}$, and the mixed strategies are distributions $\pi \in \Delta_{\mathcal{A}}$. The von Neumann winner is the minimax optimal strategy in this game. Because the game is symmetric, the von Neumann minimax theorem implies that there is always a solution $\pi$ that achieves $\mathcal{P}(\pi, \pi') \geq 0$ i.e. a single distribution $\pi$ that is preferred on average to all other distributions $\pi'$. The von Neumann winner is known to satisfy Pareto

optimality, independence of irrelevant alternatives, and population consistency (Brandl & Brandt, 2020; Brandl et al., 2016). However, a classic theorem of Gibbard (1977), extending prior work on deterministic social choice (Gibbard, 1973; Satterthwaite, 1975), implies that the von Neumann winner is not strategy-proof.

Our main results in this section will concern the regularized version of the von Neumann winner, which is defined using the regularized pairwise preferences $\widetilde{F}_{vNw}(\mathcal{A}, \mathcal{H}) = \widetilde{\mathcal{P}}_{\mathcal{H}}|_{\mathcal{A}}$. As above, the maximal distributions for $\widetilde{F}_{vNw}(\mathcal{A}, \mathcal{H})$ are given by

$$\pi^* \in \underset{\pi \in \Delta_{\mathcal{A}}}{\arg\max} \ \underset{\pi' \in \Delta_{\mathcal{A}}}{\min} \ \widetilde{\mathcal{P}}_{\mathcal{H}}(\pi, \pi'). \tag{2}$$

Due to convexity of the KL-divergence, the preference relation $\widetilde{\mathcal{P}}$ induces a convex-concave game which has a unique Nash equilibrium (for proof see Munos et al. (2024)). As above, symmetry of the game or equivalently anti-symmetry of $\widetilde{\mathcal{P}}(\pi, \pi')$ implies that the minimax value of the game, and hence the value achieved by $\pi$, is zero. We summarize these facts in the following proposition.

**Proposition 5.1.** *There is a unique maximal distribution $\pi \in \Delta\mathcal{A}$ for preference relation $\widetilde{F}_{vNw}(\mathcal{A}, \mathcal{H})$ satisfying $\widetilde{\mathcal{P}}_{\mathcal{H}}(\pi, \pi') \geq 0$ for all distributions $\pi' \in \Delta_{\mathcal{A}}$, with equality holding only when $\pi' = \pi$.*

### 5.1 STABILITY OF THE REGULARIZED VON NEUMANN WINNER

Our analysis of axioms for the regularized von Neumann winner in the subsequent section will rely on stability properties of the regularized game. In particular, we will show that the maximal distribution of the regularized von Neumann winner rule is stable under perturbations that preserve the concavity/convexity properties of the preference relation. To begin, we recall the fact that the KL-divergence is strongly convex in its first argument with respect to the $\ell_1$-norm.

**Lemma 5.2** (*Strong Convexity of KL-divergence*). *The KL-divergence is $1$-strongly convex in its first argument with respect to the $\ell_1$-norm. That is,*

$$\langle \nabla_{\pi} D_{\mathrm{KL}}(\pi \| \mu) - \nabla_{\pi'} D_{\mathrm{KL}}(\pi' \| \mu), \pi - \pi' \rangle \geq \|\pi - \pi'\|_1^2$$

The proof of Lemma 5.2 is provided in Section A. We rely on strong convexity of the KL-regularizer to prove that the regularized preference relation is stable under perturbations. The next theorem precisely quantifies this notion of stability.

**Theorem 5.3** (Stability of $\widetilde{\mathcal{P}}$). *Let $H(\pi, \pi')$ be an anti-symmetric function such that $\mathcal{P}'_{\mathcal{H}}(\pi, \pi') = \widetilde{\mathcal{P}}_{\mathcal{H}}(\pi, \pi') - H(\pi, \pi')$ is concave in $\pi$ and convex in $\pi'$. Let $\pi^*$ be maximal for $\widetilde{F}_{vNw}(\mathcal{A}, \mathcal{H})$ and let $\pi = \arg\max_{\pi_1} \min_{\pi_2} \mathcal{P}'_{\mathcal{H}}(\pi_1, \pi_2)$. Then $\|\pi^* - \pi\|_1 \leq \frac{1}{4\tau} \|\nabla_{\pi} H(\pi, \pi')|_{\pi=\pi'}\|_{\infty}$.*

The proof of Theorem 5.3 is provided in Section A. Intuitively, Theorem 5.3 states that a bounded perturbation to the preferences will result in a bounded change to the output of the perturbed rule. We now proceed to use this result to show that strategic manipulation by a small subset of raters cannot lead to significant gains.

### 5.2 REGULARIZED VON NEUMANN WINNER IS APPROXIMATELY STRATEGY-PROOF

Our first main result in this section proves that adding regularization causes the von Neumann winner rule to become approximately strategy-proof whenever the number of raters $n$ is large. In particular, the individual gains that a set of $k$ raters can achieve by strategically misreporting their preferences is at most $\frac{k}{2\tau n}$.

**Theorem 5.4** (*Approximate Strategy-Proofness of Regularized von Neuman Winner*). *Let $\tau > 0$ be the KL-regularization parameter and let $k \leq n$. Then $\widetilde{F}_{vNw}$ is $(k, \epsilon)$-approximately strategy proof for $\epsilon = \frac{k}{2\tau n}$.*

We are now ready to prove that the regularized von Neumann winner is approximately strategy proof.

*Proof of Theorem 5.4.* Let $M$ be an $|\mathcal{A}| \times |\mathcal{A}|$ matrix where $M_{a,b} = \mathcal{P}_{\mathcal{H}}(a, b)$.

Next consider the preference relation $\mathcal{P}_{\mathcal{H}'}$, where $\mathcal{H}'$ is given by letting a subset $H_*$ of $k$ raters replacing their preferences $\succ_{h_*}$ with some alternative set of preferences $\succ_{h'_*}$ denoted by $\mathcal{H}'_*$. Define $M'$ to be the matrix given by $M'_{a,b} = \mathcal{P}_{\mathcal{H}'}(a,b)$. Since $\mathcal{P}_{\mathcal{H}}$ and $\mathcal{P}_{\mathcal{H}'}$ are both averages over $n$ raters' preferences, and the only differences occur in the $k/n$ fraction of raters in $\mathcal{H}_*$, we have that for all $a, b$,

$$|M_{a,b} - M'_{a,b}| = \left| \mathop{\mathbb{E}}_{h \sim \mathcal{H}}[\mathcal{P}_h(a,b)] - \mathop{\mathbb{E}}_{h \sim \mathcal{H}'}[\mathcal{P}_h(a,b)] \right| = \left| \frac{k}{n}\mathcal{P}_{\mathcal{H}_*}(a,b) - \frac{k}{n}\mathcal{P}_{\mathcal{H}'_*}(a,b) \right| \leq \frac{2k}{n}. \quad (3)$$

Now observe that $\widetilde{\mathcal{P}}_{\mathcal{H}'}(\pi, \pi') = \widetilde{\mathcal{P}}_{\mathcal{H}}(\pi, \pi') - \pi^\top (M - M')\pi'$. The function given by

$$H(\pi, \pi') = \pi^\top (M - M')\pi'$$

is anti-symmetric because both $M$ and $M'$ are. Further $\widetilde{\mathcal{P}}_{\mathcal{H}'}(\pi, \pi')$ is concave in $\pi$ and convex in $\pi'$. Hence, if we let $\pi^*$ be maximal for $\widetilde{F}_{\text{vNw}}(\mathcal{A}, \mathcal{H})$ and $\pi$ be maximal for $\widetilde{F}_{\text{vNw}}(\mathcal{A}, \mathcal{H}')$ we can apply Theorem 5.3 followed by (3) to conclude that

$$\|\pi^* - \pi\|_1 \leq \frac{1}{4\tau}\|\nabla_\pi H(\pi, \pi')|_{\pi'=\pi}\|_\infty \leq \frac{1}{4\tau}\max_{a,b}|M_{a,b} - M'_{a,b}| \leq \frac{k}{2\tau n}. \quad (4)$$

Next we show a bound on the gain for $\mathcal{H}_*$ when switching from $\pi^*$ to $\pi$ in terms of the total variation distance,

$$\mathcal{P}_{\mathcal{H}_*}(\pi, \pi^*) = \mathop{\mathbb{E}}_{h \sim \mathcal{H}_*}\left[\Pr_{\substack{a \sim \pi \\ b \sim \pi^*}}[a \succ_h b]\right] - \mathop{\mathbb{E}}_{h \sim \mathcal{H}_*}\left[\Pr_{\substack{a \sim \pi \\ b \sim \pi^*}}[b \succ_h a]\right]$$

$$\leq \mathop{\mathbb{E}}_{h \sim \mathcal{H}_*}\left[d_{\text{TV}}(\pi, \pi^*) + \Pr_{\substack{a \sim \pi \\ b \sim \pi}}[a \succ_h b]\right] - \mathop{\mathbb{E}}_{h \sim \mathcal{H}_*}\left[\Pr_{\substack{a \sim \pi \\ b \sim \pi}}[b \succ_h a] - d_{\text{TV}}(\pi, \pi^*)\right]$$

$$= 2d_{\text{TV}}(\pi, \pi^*) + \mathcal{P}_{\mathcal{H}_*}(\pi, \pi) \leq 2d_{\text{TV}}(\pi, \pi^*)$$

where the first inequality follows from the definition of the total variation distance $d_{\text{TV}}$, and the final inequality follows from anti-symmetry of $\mathcal{P}_{\mathcal{H}_*}$. Finally by (4) we conclude

$$\mathcal{P}_{\mathcal{H}_*}(\pi, \pi^*) \leq 2d_{\text{TV}}(\pi, \pi^*) = \|\pi - \pi^*\|_1 \leq \frac{k}{2\tau n}.$$

Thus, if $k$ raters strategically misreport their preferences, then their gain is at most $\epsilon = \frac{k}{2\tau n}$. $\qquad \square$

Our next result shows that the level of approximate strategy-proofness obtained by Theorem 5.4 is a significant improvement over the unregularized version of the rule. In fact, we show that strategic misreporting by just $1$ out of $n$ raters can cause the unregularized rule to switch to outputting a policy that that rater prefers with constant (i.e. independent of $n$) probability. Our result is based on modifying and expanding a construction from prior work of Aziz et al. (2014), which showed that the von Neumann winner rule (referred to in their paper as maximal lotteries) is not $(1, 0)$-strategy proof, when $n = 5$. That is, strategic manipulation by $1$ out of $5$ raters can benefit that one rater. In contrast, we are interested in the setting where $n$ grows large, but still a single rater can significantly impact the output of the rule for their own gain.

**Theorem 5.5** (*Asymptotic Non-Strategy-Proofness of Unregularized von Neumann Winner Rule*). *For all natural numbers $n_0$, there is an $n > n_0$ such that the rule $F_{\text{vNw}}$ applied on sets of $n$ raters is not $(1, \epsilon)$-approximately strategy-proof for any $\epsilon < 4/15$.*

The proof of Theorem 5.5 is provided in the appendix Section A. The combined results of Theorem 5.4 and Theorem 5.5 establish that regularization confers the property of strategy-proofness upon the von Neumann winner rule.

## 5.3 REGULARIZED VON NEUMANN WINNER MAINTAINS OTHER AXIOMS

We now turn to the question of how regularization affects the axioms that are already satisfied by the unregularized von Neumann winner. The standard von Neumann winner satisfies independence of irrelevant alternatives, population consistency, and Pareto optimality. We will show that these axioms are also satisfied by the regularized von Neumann winner, with the single caveat that one must consider a regularized variant of Pareto optimality.

**Independence of irrelevant alternatives.**   We begin with independence of irrelevant alternatives, which states that the output of the rule should be the same, regardless of the preferences over independent alternatives that are not chosen. In the alignment setting, independence of irrelevant alternatives seems likely to be quite desirable. Consider the setting of deciding whether or not to include some additional alternative outputs judged by human raters in a preference dataset. If the alignment rule is not independent of irrelevant alternatives, then the final policy output may change significantly, even when the newly included alternatives were generally dispreferred by the raters compared to those already in the dataset. Hence failing to satisfy this axiom makes the question of how to compose and curate an alignment dataset much more difficult, whereas a rule that does satisfy the axiom would simply be unaffected by the addition of irrelevant alternatives.

**Proposition 5.6.** *The regularized von Neumann winner satisfies independence of irrelevant alternatives.*

The proof of Proposition 5.6 is provided in Section A.

**Population consistency.**   We next consider population consistency, which intuitively implies $\pi$ is maximal for the rule when combining two distinct subpopulations of raters that each individually would rate $\pi$ as maximal.

**Proposition 5.7.** *The regularized von Neumann winner satisfies population consistency.*

The proof of Proposition 5.7 is provided in Section A.

**Pareto Optimality**   Finally, we prove that the regularized von Neumann winner satisfies a regularized variant of Pareto optimality. The standard definition requires that if $\mathcal{R}$ is the output of the rule, then $\mathcal{P}_h(\pi, \pi') \geq 0$ for all $h$ implies $\mathcal{R}(\pi, \pi') \geq 0$. We will consider a regularized variant, where instead we require that $\widetilde{\mathcal{P}}_h(\pi, \pi') \geq 0$ for all $h$ implies $\mathcal{R}(\pi, \pi') \geq 0$.

**Proposition 5.8.** *The regularized von Neumann winner satisfies regularized Pareto optimality, meaning that if $\mathcal{R} = \widetilde{F}_{vNw}(\mathcal{A}, \mathcal{H})$ then $\widetilde{\mathcal{P}}_h(\pi, \pi') \geq 0$ for all $h$ implies $\mathcal{R}(\pi, \pi') \geq 0$.*

The proof of Proposition 5.8 is provided in Section A. Overall, the results in this section show that regularization substantially improves startegy-proofness, while maintaining other beneficial axiomatic properties of the von Neumann winner rule.

## 6 THE BORDA COUNT RULE

In this section we analyze the axiomatic properties of the regularized Borda count rule. This rule is of particular importance in the RLHF setting, where it has been shown that the standard methods based on the Bradely-Terry model maximize a reward that produces precisely the same rankings as the Borda count (Siththaranjan et al., 2024). We begin with the standard definition of the Borda count rule. Given a set of $\mathcal{H}$ of raters and a set $\mathcal{A}$ of $n$ alternatives, for each rater $h \in \mathcal{H}$ and alternative $a \in \mathcal{A}$, assign the score $s_h(a) = n - k$ if $a$ is ranked in the $k$-th position by $h$ i.e. the top ranked alternative gets score $n - 1$, the second ranked gets $n - 2$ and so on. The Borda count rule then ranks the alternatives by their total score $s(a) = \sum_h s_h(a)$. For the setting of probabilistic social choice, the rule ranks alternatives by their expected total score. Formally, we define the preference relation $\mathcal{R} = F_{\text{Borda}}(\mathcal{A}, \mathcal{H})$ output by the Borda count rule by

$$\mathcal{R}(\pi, \pi') = \frac{2}{n} \left( \mathbb{E}_{a \sim \pi} \sum_{h \in \mathcal{H}} s_h(a) - \mathbb{E}_{a' \sim \pi'} \sum_{h \in \mathcal{H}} s_h(a') \right).$$

The value of $\mathcal{R}(\pi, \pi')$ is non-negative if and only if $\pi$ has a higher expected total score than $\pi'$. We have normalized the total score by $\frac{2}{n}$ above for convenience in the later analysis, and this does not affect the relative ordering of any pair of distributions under $\mathcal{R}$. Observe that the score $s_h(a)$ is equal to the number of alternatives $b$ such that $a \succ_h b$. That is

$$s_h(a) = \sum_{b \in \mathcal{A}} \frac{1 + \mathcal{P}_h(a, b)}{2} = \frac{n}{2} + \frac{1}{2} \sum_{b \in \mathcal{A}} \mathcal{P}_h(a, b)$$

Thus the relation $\mathcal{R}$ output by the Borda count rule can equivalently be expressed in the form

$$\mathcal{R}(\pi, \pi') = \mathbb{E}_{a \sim \pi} \mathbb{E}_{h \sim \mathcal{H}} \sum_{b \in \mathcal{A}} \mathcal{P}_h(a, b) - \mathbb{E}_{a' \sim \pi'} \mathbb{E}_{h \sim \mathcal{H}} \sum_{b' \in \mathcal{A}} \mathcal{P}_h(a', b')$$

$$= \mathbb{E}_{a \sim \pi} \sum_{b \in \mathcal{A}} \mathcal{P}_{\mathcal{H}}(a, b) - \mathbb{E}_{a' \sim \pi'} \sum_{b' \in \mathcal{A}} \mathcal{P}_{\mathcal{H}}(a', b'). \tag{5}$$

Note that this form depends only on pairwise comparisons between alternatives.

In practical RLHF the reward function $r(a)$ for $a \in \mathcal{A}$ produced via training with the Bradley-Terry model yields the same ranking over alternatives as the Borda count. However, this reward function is not numerically equal to the Borda score. Hence, to appropriately model practical RLHF, we will be interested in any relation $\mathcal{Q}$ derived from a reward $r$ which produces the same rankings over alternatives as the Borda score $s(a)$. That is, we consider all $\mathcal{Q}$ such that

$$\mathcal{Q}(\pi, \pi') = \mathbb{E}_{a \sim \pi} r(a) - \mathbb{E}_{a' \sim \pi'} r(a')$$

where $r(a) \geq r(b)$ if and only if $s(a) \geq s(b)$.

As before, we will be interested in a regularized version of the Borda count rule, as this matches the standard practice in RLHF. In particular, the standard RLHF algorithm will output a policy $\pi$ that maximizes the Borda count score plus a KL-divergence regularization term. Hence, we define the output of the regularized Borda count rule $\widetilde{\mathcal{R}} = \widetilde{F}_{\text{Borda}}(\mathcal{A}, \mathcal{H})$ by

$$\widetilde{\mathcal{R}}(\pi, \pi') = \mathbb{E}_{a \sim \pi} \sum_{b \in \mathcal{A}} \mathcal{P}_{\mathcal{H}}(a, b) - \mathbb{E}_{a' \sim \pi'} \sum_{b' \in \mathcal{A}} \mathcal{P}_{\mathcal{H}}(a', b') - D_{\text{KL}}(\pi \| \mu) + D_{\text{KL}}(\pi' \| \mu).$$

We similarly define $\widetilde{\mathcal{Q}}(\pi, \pi') = Q(\pi, \pi') - D_{\text{KL}}(\pi \| \mu) + D_{\text{KL}}(\pi' \| \mu)$ for any relation $\mathcal{Q}(\pi, \pi')$ induced by a reward $r$ that produces the same rankings over alternatives as the Borda count.

Because the original, unregularized Borda count rule is a deterministic, non-dictatorial rule, Arrow's theorem (Arrow, 1951) implies that it must violate either Pareto optimality or independence of irrelevant alternatives. As it turns out, it violates the latter i.e. $F_{\text{Borda}}$ is not independent of irrelevant alternatives. However, as we saw for the von Neumann winner, regularization sometimes has unexpected benefits. In spite of this initial optimism, we will next prove that the regularized Borda count also violates independence of irrelevant alternatives.

**Proposition 6.1.** *Let $\mathcal{Q}$ be any preference relation which yields the same ranking over alternatives as the Borda count, and $\widetilde{\mathcal{Q}}$ be the regularized relation. For all $\epsilon < 1/2$, the alignment rule that outputs $\widetilde{\mathcal{Q}}$ is not $\epsilon$-approximately independent of irrelevant alternatives.*

The proof of Proposition 6.1 appears in Section A. Hence, the key axiom of independence of irrelevant alternatives remains unsatisfied by the regularized Borda count rule. This is unlike the case of the regularized von Neumann winner, where regularization was able to significantly improve the strategy-proofness of the rule.

## 7 EXPERIMENTS

In this section we demonstrate the strategy proofness of the regularized von Neumann rule on synthetic data. We consider the counterexample appearing in the proof of Theorem 5.5 in Section A, with $|\mathcal{A}| = 3$ alternatives and a set $\mathcal{H}$ of $n = 64$ raters. In this setting there are two preference relations $\mathcal{P}$ and $\mathcal{P}'$ that differ only in that a single rater switches their preferences from $a_1 \succ a_3 \succ a_2$ in $\mathcal{P}$ to $a_1 \succ a_2 \succ a_3$ in $\mathcal{P}'$. We compare the regularized and unregularized von Neumann winner

on the two preference relations $\mathcal{P}$ and $\mathcal{P}'$. We implement the Nash MD algorithm of Munos et al. (2024) to compute the maximal policy given by the regularized von Neumann winner rule with regularization parameter $\tau = 0.1$. For the unregularized von Neumann winner, we implement the self-play preference optimization (SPO) algorithm of Swamy et al. (2024). Both algorithms are initialized with a policy $\mu$ that assigns probability 0.35, 0.21, and 0.43 to $a_1, a_2$, and $a_3$ respectively. The distribution $\mu$ also serves as the regularizer for the regularized von Neumann winner rule. In Figure 1 we then plot for each algorithm the probability assigned to each alternative $a_1, a_2, a_3$ as function of training steps. The results demonstrate that a strategic change in reported preferences in this setting has a minor impact on the regularized von Neumann winner, but a results in a very large change for the output of the unregularized rule. Notably, this change dramatically increases the probability of outputting $a_1$, which is the most preferred option of the single, strategically misreporting rater.

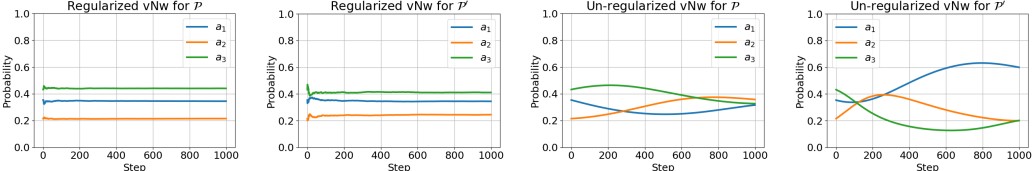

Figure 1: Regularized and un-regularized von Neumann winner policies throughout training for two sets of rater preferences $\mathcal{P}$ and $\mathcal{P}'$ that only differ due to strategic misreporting by a single rater who most prefers $a_1$.

## 8    DISCUSSION

We have shown that regularization to a reference policy makes the von Neumann winner rule approximately strategy-proof, while preserving other key social choice axioms. This is a clear advantage over the Borda count rule implemented by standard RLHF, which is not independent of irrelevant alternatives in its regularized or unregularized variants. We next discuss limitations and potential future directions of research.

On the limitations side, our results show that the regularized von Neumann winner only satisfies a regularized version of Pareto efficiency. This is in fact necessary for any regularized rule, as the reference policy may assign arbitrarily small probability to a particular outcome which all raters may have as their favorite. This also points to an interesting avenue for future research, where additional assumptions on the reference policy might be used to recover (approximate) Pareto efficiency. Another avenue for future theoretical work is to determine if there are additional probabilistic social choice rules that interact positively with KL-regularization. Such rules could inform the design of future alignment algorithms that could potentially improve upon the axiomatic properties of the von Neumann winner.

## 9 REPRODUCIBILITY STATEMENT

All assumptions and formal definitions for the theoretical results are included in the main body of the paper. All proofs missing from the main body are included in Section A.

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

## A    MISSING PROOFS FOR SECTION 5

In this section we provide proofs of the results regarding stability of the regularized von Neumann rule from Section 5.

We begin with Lemma 5.2.

*Proof of Lemma 5.2.* Let $H(\pi) = -\sum_x \pi(x) \log \pi(x)$ denote the entropy of $\pi$. By Pinsker's inequality $D_{\mathrm{KL}}(\pi, \mu) \geq \frac{1}{2}\|\pi - \mu\|_1^2$

$$\frac{1}{2}\|\pi - \mu\|_1^2 \leq D_{\mathrm{KL}}(\pi, \mu) = -H(\pi) + H(\mu) + \langle -\nabla_\mu H(\mu), \pi - \mu \rangle$$

Hence the negative entropy is 1-strongly convex with respect to the $\ell_1$-norm. Observe that

$$D_{\mathrm{KL}}(\pi, \mu) = \sum_x \pi(x) \log\left(\frac{\pi(x)}{\mu(x)}\right) = -H(\pi) - \sum_x \pi(x) \log(\mu(x)).$$

The first term is 1-strongly convex with respect to the $\ell_1$-norm and the second term is linear. Hence $D_{\mathrm{KL}}(\pi, \mu)$ is 1-strongly convex with respect to the $\ell_1$-norm. $\square$

We next turn to the proof of Theorem 5.3, that shows that the regularized von Neumann rule is stable under sufficiently bounded perturbations to the regularized preferences.

*Proof of Theorem 5.3.* Let $M$ be an $|\mathcal{A}| \times |\mathcal{A}|$ matrix where $M_{a,b} = \mathcal{P}_\mathcal{H}(a, b)$. Then we can rexpress the von Neumann winner as the equilibrium in the symmetric two-player game with payoffs for $\pi$ and $\pi'$ given respectively by

$$R_\mathcal{H}(\pi, \pi') = \pi^\top M \pi' - \tau D_{\mathrm{KL}}(\pi \| \mu) \qquad R_\mathcal{H}(\pi', \pi) = {\pi'}^\top M \pi - \tau D_{\mathrm{KL}}(\pi' \| \mu)$$

The payoffs to each player are concave so the game has a Nash equilibrium by Theorem 1 of Rosen (1965). Let $\vec{\pi} = (\pi, \pi')$ and define $G(\vec{\pi}) = (-\nabla_\pi R(\pi, \pi'), -\nabla_{\pi'} R(\pi', \pi))$. Then Rosen

(1965) shows that if $\vec{\pi}^* = (\pi^*, {\pi'}^*)$ is an equilibrium of the game, then for any pair of distributions $\vec{\eta} = (\eta, \eta')$ we have

$$\langle G(\vec{\pi}^*), \vec{\eta} - \vec{\pi}^* \rangle \geq 0. \tag{6}$$

Next consider the perturbed preference relation $\mathcal{P}'_{\mathcal{H}}(\pi, \pi') = \widetilde{\mathcal{P}}_{\mathcal{H}}(\pi, \pi') - H(\pi, \pi')$, where by the assumptions of the theorem we have that $\mathcal{P}'_{\mathcal{H}}(\pi, \pi')$ is concave in $\pi$ and convex in $\pi'$. As above $\mathcal{P}'_{\mathcal{H}}(\pi, \pi')$ defines a symmetric game with payoffs for $\pi$ and $\pi'$ given by

$$R'_{\mathcal{H}}(\pi, \pi') = \pi^\top M \pi' - H(\pi, \pi') - \tau D_{\mathrm{KL}}(\pi \| \mu)$$
$$R'_{\mathcal{H}}(\pi', \pi) = {\pi'}^\top M \pi - H(\pi', \pi) - \tau D_{\mathrm{KL}}(\pi' \| \mu).$$

Furthermore, the payoffs are concave for both players, so as above we have that $G'(\vec{\pi}) = (-\nabla_\pi R'_{\mathcal{H}}(\pi, \pi'), -\nabla_{\pi'} R'_{\mathcal{H}}(\pi', \pi))$ satisfies (6).

By Lemma 5.2, the negative payoff function $-R_{\mathcal{H}}(\pi, \pi')$ is $\tau$-strongly convex in $\pi$ with respect to the $\ell_1$-norm because it is the sum of a linear function of $\pi$ with the $\tau$-strongly convex function $\tau D_{\mathrm{KL}}(\pi \| \mu)$. Hence, for any $\vec{\pi} = (\pi, \pi'), \vec{\eta} = (\eta, \eta')$,

$$\langle G(\vec{\pi}) - G(\vec{\eta}), \vec{\pi} - \vec{\eta} \rangle = \langle -\nabla_\pi R(\pi, \pi'), \pi - \eta \rangle + \langle -\nabla_{\pi'} R(\pi', \pi), \pi' - \eta' \rangle$$
$$+ \langle -\nabla_\eta R(\eta, \eta'), \eta - \pi \rangle + \langle -\nabla_{\eta'} R(\eta', \eta), \eta' - \pi' \rangle$$
$$\geq 2\tau \|\pi - \eta\|_1^2 + 2\tau \|\pi' - \eta'\|_1^2 \tag{7}$$

Let $\pi^* = \widetilde{F}_{\mathrm{vNw}}(\mathcal{A}, \mathcal{H})$ and let $\pi = \arg\max_{\pi_1} \min_{\pi_2} \mathcal{P}'_{\mathcal{H}}(\pi_1, \pi_2)$. Further let $\vec{\pi}^* = (\pi^*, \pi^*)$ and $\vec{\pi} = (\pi, \pi)$.

By (7) we have

$$4\tau \|\pi - \pi^*\|_1^2 \leq \langle G(\vec{\pi}) - G(\vec{\pi}^*), \vec{\pi} - \vec{\pi}^* \rangle$$
$$= \langle G(\vec{\pi}), \vec{\pi} - \vec{\pi}^* \rangle - \langle G(\vec{\pi}^*), \vec{\pi} - \vec{\pi}^* \rangle$$
$$\leq \langle G(\vec{\pi}), \vec{\pi} - \vec{\pi}^* \rangle \qquad \text{((6) applied to } G(\vec{\pi}^*))$$
$$= \left\langle G(\vec{\pi}) - G'(\vec{\pi}) + \vec{G'(\pi)}, \vec{\pi} - \vec{\pi}^* \right\rangle$$
$$= \langle G(\vec{\pi}) - G'(\vec{\pi}), \vec{\pi} - \vec{\pi}^* \rangle + \langle G'(\vec{\pi}), \vec{\pi} - \vec{\pi}^* \rangle$$
$$\leq \langle G(\vec{\pi}) - G'(\vec{\pi}), \vec{\pi} - \vec{\pi}^* \rangle \qquad \text{((6) applied to } G'(\vec{\pi}))$$
$$\leq \|G(\vec{\pi}) - G'(\vec{\pi})\|_\infty \cdot \|\vec{\pi} - \vec{\pi}^*\|_1 \qquad \text{(Hölder's inequality)}$$
$$= \|\nabla_\pi H(\pi, \pi')|_{\pi' = \pi}\|_\infty \cdot \|\vec{\pi} - \vec{\pi}^*\|_1$$

Dividing both sides by $4\tau \|\pi - \pi^*\|_1$ yields the desired result. $\qquad \square$

Next, we turn to the proof that the standard von Neumann winner rule is not approximately strategy proof.

*Proof of Theorem 5.5.* We proceed by constructing a set of alternatives $\mathcal{A}$ and raters $\mathcal{H}$ such that there exists a subset $\mathcal{H}_*$ of $k = 1$ raters that can strategically misreport preferences to obtain constant gain. Consider a set $\mathcal{A}$ of three alternatives $a_1, a_2, a_3$. We construct 8 disjoint subsets of raters $\mathcal{H}_1, \ldots, \mathcal{H}_4$ and $\overline{\mathcal{H}}_1, \ldots, \overline{\mathcal{H}}_4$ as follows. Within each subset the preferences of the raters are identical and are given by

$$
\begin{array}{llll}
\mathcal{H}_1: & a_1 \succ a_3 \succ a_2 & \overline{\mathcal{H}}_1: & a_2 \succ a_3 \succ a_1 \\
\mathcal{H}_2: & a_1 \succ a_2 \succ a_3 & \overline{\mathcal{H}}_2: & a_3 \succ a_2 \succ a_1 \\
\mathcal{H}_3: & a_2 \succ a_3 \succ a_1 & \overline{\mathcal{H}}_3: & a_1 \succ a_3 \succ a_2 \\
\mathcal{H}_4: & a_3 \succ a_1 \succ a_2 & \overline{\mathcal{H}}_4: & a_2 \succ a_1 \succ a_3
\end{array} \tag{8}
$$

Let $n_0$ be any natural number, and set $l$ such that $n = 10l - 5 > n_0$. Let the subsets of raters have sizes $|\mathcal{H}_1| = |\mathcal{H}_2| = |\mathcal{H}_4| = l$, $|\mathcal{H}_3| = 2l$, and let $|\overline{\mathcal{H}}_1| = |\overline{\mathcal{H}}_2| = |\overline{\mathcal{H}}_4| = l - 1$, and $|\overline{\mathcal{H}}_3| = 2l - 2$.

Let $\mathcal{H} = \cup_{i=1}^{4}(\mathcal{H}_i \cup \overline{\mathcal{H}}_i)$ and observe that $|\mathcal{H}| = n$. We can now write the preference relation $\mathcal{P}_{\mathcal{H}}(a, b)$ in the form of a $3 \times 3$ matrix,

$$
\mathcal{P}_{\mathcal{H}} = \begin{pmatrix} 0 & 1/n & -1/n \\ -1/n & 0 & 1/n \\ 1/n & -1/n & 0 \end{pmatrix} \tag{9}
$$

This is just $\frac{1}{n}$ times the payoffs of rock-paper-scissors, and the unique Nash equilibrium (and hence the von Neumann winner) is given by $\pi = (1/3, 1/3, 1/3)$. However if one rater $h_*$ from $\mathcal{H}_1$ switches their preferences to $a_1 \succ a_2 \succ a_3$ to produce a manipulated set of rater preferences $\mathcal{H}'$ then the preference relation $\mathcal{P}_{\mathcal{H}'}(a, b)$ becomes

$$
\mathcal{P}_{\mathcal{H}'} = \begin{pmatrix} 0 & 1/n & -1/n \\ -1/n & 0 & 3/n \\ 1/n & -3/n & 0 \end{pmatrix} \tag{10}
$$

In this game the unique von Neumann winner is given by $\pi' = (3/5, 1/5, 1/5)$. Observe that $\pi'$ puts significantly more weight on $h_*$'s most preferred alternative. Further, direct calculation of the preference relation for $h_*$ yields

$$
\mathcal{P}_{h_*}(\pi', \pi) = \Pr_{\substack{a \sim \pi' \\ b \sim \pi}}[a \succ b] - \Pr_{\substack{a \sim \pi' \\ b \sim \pi}}[b \succ a] = 7/15 - 3/15 = 4/15
$$

which completes the proof. $\qquad\square$

We now give the proof of Proposition 5.6, that the regularized von Neumann winner is independent of irrelevant alternatives.

*Proof of Proposition 5.6.* Let $\mathcal{A} \subseteq \mathcal{B} \subseteq \mathcal{U}$, and let $\mathcal{H}, \mathcal{H}'$ be two sets of raters that have identical preferences over $\mathcal{A}$. Let $M$ be the matrix given by $M_{a,b} = \mathcal{P}_{\mathcal{H}}(a, b)$, and $M'$ the matrix given by $M'_{a,b} = \mathcal{P}_{\mathcal{H}'}(a, b)$. Let $M_{\mathcal{A}}$ be the submatrix of $M$ restricted to $\mathcal{A}$ and $M'_{\mathcal{A}}$ the submatrix of $M'$ restricted to $\mathcal{A}$. Because $\mathcal{H}$ and $\mathcal{H}'$ have identical preferences over $\mathcal{A}$, we have $M_{\mathcal{A}} = M'_{\mathcal{A}}$. Then for any $\pi, \pi'$ supported on $\mathcal{A}$ we have

$$
\begin{aligned}
\widetilde{\mathcal{P}}_{\mathcal{H}}(\pi, \pi') &= \pi^\top M \pi' - \tau D_{\mathrm{KL}}(\pi \| \mu) + \tau D_{\mathrm{KL}}(\pi' \| \mu) \\
&= \pi^\top M'_{\mathcal{A}} \pi' - \tau D_{\mathrm{KL}}(\pi \| \mu) + \tau D_{\mathrm{KL}}(\pi' \| \mu) \qquad (\pi, \pi' \text{ are supported on } \mathcal{A}) \\
&= \widetilde{\mathcal{P}}_{\mathcal{H}'}(\pi, \pi')
\end{aligned}
$$

Hence $\widetilde{F}_{\mathrm{vNw}}(\mathcal{B}, \mathcal{H})|_A = \widetilde{\mathcal{P}}_{\mathcal{H}}|_A = \widetilde{\mathcal{P}}_{\mathcal{H}'}|_A = \widetilde{F}_{\mathrm{vNw}}(\mathcal{B}, \mathcal{H}')$ i.e. $\widetilde{F}_{\mathrm{vNw}}$ is independent of irrelevant alternatives. $\qquad\square$

Next we provide the proof of Proposition 5.7, demonstrating that the regularized von Neumann winner satisfies population consistency.

*Proof of Proposition 5.7.* Let $\mathcal{H}$ and $\mathcal{H}'$ be two disjoint sets of raters, and suppose that $\pi$ is maximal for both $\widetilde{F}_{\mathrm{vNw}}(\mathcal{A}, \mathcal{H})$ and $\widetilde{F}_{\mathrm{vNw}}(\mathcal{A}, \mathcal{H}')$. Let

$$
\lambda = \frac{|\mathcal{H}|}{|\mathcal{H}| + |\mathcal{H}'|}.
$$

Further let $M$ be the matrix given by $M_{a,b} = \mathcal{P}_{\mathcal{H}}(a, b)$ and $M'$ be the matrix given by $M'_{a,b} = \mathcal{P}_{\mathcal{H}}(a, b)$. Then $\mathcal{P}_{\mathcal{H} \cup \mathcal{H}'}(a, b) = \lambda M_{a,b} + (1 - \lambda) M'_{a,b}$. Hence, for any $\pi' \in \Delta_{\mathcal{A}}$ we have

$$
\begin{aligned}
\widetilde{\mathcal{P}}_{\mathcal{H} \cup \mathcal{H}'}(\pi, \pi') &= \pi^\top (\lambda M + (1 - \lambda) M') \pi - \tau D_{\mathrm{KL}}(\pi \| \mu) + \tau D_{\mathrm{KL}}(\pi' \| \mu) \\
&= \lambda(\pi^\top M \pi' - \tau D_{\mathrm{KL}}(\pi \| \mu) + \tau D_{\mathrm{KL}}(\pi' \| \mu)) \\
&\quad + (1 - \lambda)(\pi^\top M' \pi' - \tau D_{\mathrm{KL}}(\pi \| \mu) + \tau D_{\mathrm{KL}}(\pi' \| \mu)) \\
&= \lambda \widetilde{\mathcal{P}}_{\mathcal{H}}(\pi, \pi') + (1 - \lambda) \widetilde{\mathcal{P}}_{\mathcal{H}'}(\pi, \pi') \geq 0
\end{aligned}
$$

where the final inequality follows from Proposition 5.1. Thus $\widetilde{\mathcal{P}}(\pi, \pi') \geq 0$ for all $\pi'$ i.e. $\pi$ is maximal for $\widetilde{F}_{\mathrm{vNw}}(\mathcal{A}, \mathcal{H} \cup \mathcal{H}')$. $\qquad\square$

Next, we prove Proposition 5.8, showing the regularized von Neumann winner rule satisfies regularized Pareto optimality.

*Proof of Proposition 5.8.* Suppose that $\pi, \pi' \in \Delta\mathcal{A}$ satisfy $\widetilde{\mathcal{P}}_h(\pi, \pi') \geq 0$ for all $h$. Since $\mathcal{R} = \widetilde{\mathcal{P}}_{\mathcal{H}}$ is the output of the von Neumann winner rule $\widetilde{F}_{\text{vNw}}$ we have

$$\widetilde{\mathcal{P}}_{\mathcal{H}}(\pi, \pi') = \mathop{\mathbb{E}}_{h \sim \mathcal{H}}[\widetilde{\mathcal{P}}_h(\pi, \pi')] \geq 0$$

implying that the von Neumann winner satisfies Pareto optimality. $\square$

Finally, we prove Proposition 6.1 showing that the regularized Borda count rule does not satisfy approximate independence of irrelevant alternatives.

*Proof of Proposition 6.1.* We construct a counterexample with three alternatives and two raters for which regularized Borda count violates independence of irrelevant alternatives. Let $\mathcal{A} = \{a_1, a_2\} \subseteq \mathcal{B} = \{a_1, a_2, a_3\}$ and let there be two raters $\mathcal{H} = \{h_1, h_2\}$ with preferences

$$a_1 \succ_{h_1} a_2 \succ_{h_1} a_3$$
$$a_2 \succ_{h_2} a_3 \succ_{h_2} a_1$$

Let $s(a) = \sum_{b \in \mathcal{A}} \mathcal{P}_{\mathcal{H}}(a, b)$. Then $s(a_1) = 2$, $s(a_2) = 3$, and $s(a_3) = 1$. Consider also the alternative set of ratings $\mathcal{H}' = \{h_1', h_2'\}$ with preferences

$$a_1 \succ_{h_1} a_3 \succ_{h_1} a_2$$
$$a_2 \succ_{h_2} a_1 \succ_{h_2} a_3$$

Observe that $\mathcal{H}$ and $\mathcal{H}'$ have identical preferences over $a_1, a_2$. Letting $s'(a) = \sum_{b \in \mathcal{A}} \mathcal{P}_{\mathcal{H}'}(a, b)$, we then have $s'(a_1) = 3$, $s'(a_2) = 2$, and $s'(a_3) = 1$.

Next let the reference measure be uniform i.e. $\mu(a_1) = \mu(a_2) = \mu(a_3) = 1/3$. Further consider the two distributions $\pi = \mathbf{1}_{a_1}$ and $\pi' = \mathbf{1}_{a_2}$ that put all their mass on $a_1$ and $a_2$ respectively. Hence for any reward $r$ that produces the same rankings as the Borda score $s$ we have $r(a_1) < r(a_2)$, and for any reward $r'$ that produces the same rankings as the Borda score $s'$ we have $r'(a_1) \geq r'(a_2)$. Next let $\mathcal{Q}$ be the relation induced by $r$, and $\widetilde{\mathcal{Q}}$ be its regularized version. Then we have

$$\widetilde{\mathcal{Q}}(\pi, \pi') = \mathcal{Q}(\pi, \pi') - D_{\text{KL}}(\pi \| \mu) + D_{\text{KL}}(\pi' \| \mu)$$
$$= r(a_1) - r(a_2) - \log(3) + \log(3) = r(a_1) - r(a_2) < 0$$

where the last line follows because $s(a_1) < s(a_2)$ which implies that $r(a_1) < r(a_2)$ as they produce the same rankings. Note that for any distributions $\hat{\pi}, \hat{\pi}'$ satisfying $d_{\text{TV}}(\hat{\pi}, \pi) < 1/2$ and $d_{\text{TV}}(\hat{\pi}', \pi') < 1/2$ we have $\hat{\pi}(a_1) > 1/2$, $\hat{\pi}(a_2) < 1/2$, $\hat{\pi}'(a_1) < 1/2$, and $\hat{\pi}'(a_2) > 1/2$. Therefore, for any relation $\mathcal{Q}'$ induced by the reward $r'$ we have

$$\widetilde{\mathcal{Q}}'(\pi, \pi') = \mathcal{Q}'(\pi, \pi') - D_{\text{KL}}(\pi \| \mu) + D_{\text{KL}}(\pi' \| \mu)$$
$$= \mathop{\mathbb{E}}_{a \sim \hat{\pi}} r'(a) - \mathop{\mathbb{E}}_{a' \sim \hat{\pi}'} r'(a') - \log(3) + \log(3)$$
$$= \hat{\pi}(a_1) r'(a_1) + \hat{\pi}(a_2) r'(a_2) - \hat{\pi}'(a_1) r(a_1) - \hat{\pi}'(a_2) r'(a_2)$$
$$= r'(a_1) (\underbrace{\hat{\pi}(a_1)}_{>1/2} - \underbrace{\hat{\pi}'(a_1)}_{<1/2}) - r'(a_2) (\underbrace{\hat{\pi}'(a_2)}_{<1/2} - \underbrace{\hat{\pi}(a_2)}_{>1/2})$$
$$\geq 0$$

where the last line follows from the total variation bounds on $\hat{\pi}, \hat{\pi}'$ and because $r'(a_1) \geq r'(a_2)$. Thus, even though $\mathcal{H}$ and $\mathcal{H}'$ have identical preferences over $a_1, a_2$, the relative order under $\widetilde{\mathcal{Q}}$ and $\widetilde{\mathcal{Q}}'$ is swapped, violating the $\epsilon$-approximate independence of irrelevant alternatives for any $\epsilon < 1/2$. $\square$

