# OpenReview forum: "Provable Guarantees from Practical Regularization for Alignment with Human Preferences"
_ICLR.cc/2026/Conference — Submitted to ICLR 2026_

### Official Review · Reviewer_FGJM · 2025-10-29

**Soundness:** 2
**Presentation:** 1
**Contribution:** 2
**Rating:** 2
**Confidence:** 3

**Summary:**

The paper studies alignment through probabilistic social choice by inserting a KL term with respect to a reference policy into a pairwise preference relation. For the von Neumann winner (maximal lotteries), the authors optimize a maximin objective over distributions; for Borda, they define a Borda score plus KL relation. They report that the KL-regularized von Neumann winner admits a unique solution, satisfies independence of irrelevant alternatives and population consistency, and is approximately strategy-proof with coalition gain bounded on the order of $1/n$. Pareto is obtained only in a regularized sense due to the reference policy. For the Borda variant, the authors show independence of irrelevant alternatives fails even with KL.

**Strengths:**

The paper tackles a well motivated question about understanding alignment from a social choice perspective and in particular focuses on the impact of KL divergence.

**Weaknesses:**

### Scope compared to alignment

The first thing I want to raise is the difficulty of positioning this paper within the broader landscape of alignment and social choice. Below I’ve detailed a few points. I think the paper would be stronger if these were at least addressed or mentioned in the main text


The paper analyzes a maximin game over distributions with KL inside the game for the vNW rule, and a Borda score plus KL objective for the Borda section. This is not the mainstream “maximize expected learned reward subject to a KL penalty to a reference policy” used in standard RLHF. The distinction matters because the pipeline, guarantees, and failure modes differ.
Moreover, the modeling fixes one candidate set $A$ and studies a single game over $\Delta_A$. The paper could have benefited from clearly stating this difference up front and framing all connections to Bradley–Terry and Borda as aggregation properties rather than training-objective equivalences.
Alignment in practice trains a context-conditioned policy with a KL term defined per prompt, not a single global lottery over a static $A$. It neither models the per-prompt ballots that generate comparisons nor the outer expectation over prompts that determines reward and policy updates in practice. Section 6 explains that Bradley–Terry based methods “produce the same rankings as Borda,” then defines a regularized Borda preference over distributions. That analysis ranks alternatives after aggregating pairwise data and is not the same as optimizing the PPO-style RLHF objective with a learned reward and a per-token KL penalty.
Strategy-proofness is presented as central, but the motivation for RLHF is thin. The paper does not explain when real raters would have incentives or the ability to coordinate misreports in modern preference data collection, nor how this metric maps to downstream harms in model behavior.

### Limited technical depth and novelty

The main proofs lean on strong convexity on the simplex, which yields stability of the unique equilibrium under small perturbations. These arguments are correct but routine and do not appear to deliver tight constants.

### Axioms applied to a regularized individual relation are normatively odd

The definition of $\tilde{P}_h$ makes each rater’s “preference” depend on the reference policy $\mu$ rather than only on their ranking over alternatives. This departs from the standard unanimity idea that aggregates human preferences rather than mixing them with a modeling prior. That means the rule can rank $\pi$ over $\pi'$ even when all raters prefer $\pi'$ in the ordinary sense, as long as $\pi$ is closer to $\mu$. If alignment is meant to elicit preferences independent of the current policy, this redefinition needs a stronger justification than is provided.

### Typos and notation issues

The paper is hard to read and contains many notational inconsistencies and typos. For example:
• The paper alternates between “strategy-proof,” “strategy proof,” and “strategy proofness.”

• $n$ denotes the number of raters in Sections 3–5 and the number of alternatives in Section 6.

• Inconsistent notation for $D_{\mathrm{KL}}$.

• Spelling mistakes at lines 129, 299, 400, 457.

**Questions:**

Please address my concerns above.

---

> ### Author Response · Authors · 2025-11-25
>
> We are glad to see that you found our paper tackling a well motivated question about understanding alignment from a social choice perspective and in particular focuses on the impact of KL divergence.
>
> 1. *“It neither models the per-prompt ballots that generate comparisons nor the outer expectation over prompts”*
>
> We could of course define the preferences $\mathcal{P}$ to be prompt dependent $\mathcal{P}\_{x}$, and add an outer expectation over prompts throughout the paper, and all the results would remain the same. In our opinion, this would add significant notational complexity and obscure the main mathematical properties of pluralistic alignment that we are trying to illustrate. The decision to leave out the outer expectation over prompts is also a common one that has appeared in prior work that studies alignment through the lens of social choice theory [1].
>
> [1] Ge et al. Axioms for AI Alignment from Human Feedback. NeurIPS 2024.
>
> 2. *“Section 6 explains that Bradley–Terry based methods “produce the same rankings as Borda,” then defines a regularized Borda preference over distributions. That analysis ranks alternatives after aggregating pairwise data and is not the same as optimizing the PPO-style RLHF objective with a learned reward and a per-token KL penalty.”*
>
> As was shown originally in [2], the optimal learned reward function used in standard Bradley-Terry based RLHF ranks alternatives in the same order as the Borda count. Thus, if one maximizes the learned reward (via PPO or any other method) it is equivalent to view this as maximizing the Borda count score.
>
> Perhaps your concern is that adding the KL-regularizer directly to the Borda score may be different than adding it to the learned rewards, as they produce the same ranking but are not numerically equivalent objective functions? This is also not an issue for our results, as Proposition 6.1 applies equally well to KL-regularization of **any reward function** that produces the same rankings as the Borda count. We have updated the paper to make this point clear.
>
> [2] Siththaranjan et al. Distributional preference learning: Understanding and accounting for hidden context in RLHF. ICLR 2024.
>
> 3. *“Strategy-proofness is presented as central, but the motivation for RLHF is thin. The paper does not explain when real raters would have incentives or the ability to coordinate misreports in modern preference data collection, nor how this metric maps to downstream harms in model behavior.”*
>
> Theorem 5.5 shows that, for the unregularized von Neumann winner, even a single rater can cause a large change in the policy via misreporting. Hence, the unregularized rule is susceptible to misreporting even without coordination among raters.
>
> As for incentives for misreporting, modern language models are more and more frequently used to generate pieces of code both in open source products and in industry. If an attacker wanted to increase the likelihood of generated code containing a certain type of vulnerability, they could attempt to be selected as a rater and then misreport their preferences in order to introduce such vulnerabilities wherever the LLM is used. If a single rater or small group of raters is able to have this effect on an LLM, this would clearly be an attack vector that bad actors would be highly incentivized to use.
>
> 4. *“The definition of $\widetilde{\mathcal{P}}_h$ makes each rater’s “preference” depend on the reference policy $\mu$ rather than only on their ranking over alternatives. This departs from the standard unanimity idea that aggregates human preferences rather than mixing them with a modeling prior. That means the rule can rank $\pi$ over $\pi’$ even when all raters prefer $\pi’$ in the ordinary sense, as long as $\pi$ is closer to $\mu$. If alignment is meant to elicit preferences independent of the current policy, this redefinition needs a stronger justification than is provided.”*
>
> In the typical RLHF setting, the reference policy $\mu$ is the pretrained and SFTed model. Pretraining is typically done over massive human-generated text datasets, and so the reference model $\mu$ can be thought of as being a proxy for the “uniform distribution on human generated text.” In particular, we can think of $\mu$ as an initial, unbiased distribution on language, with future RLHF training introducing possible biases for specific contexts, based on human preferences in those contexts. In this setting, regularization to the reference policy $\mu$ is natural in that we want to satisfy human preferences as much as possible, while avoiding introducing huge changes to the distribution. From the social choice point of view, this can be thought of as introducing a soft constraint on the set of candidates/alternatives, based on some agreed-upon prior.

---

### Official Review · Reviewer_fHB9 · 2025-10-29

**Soundness:** 3
**Presentation:** 3
**Contribution:** 3
**Rating:** 6
**Confidence:** 3

**Summary:**

This paper analyzes AI alignment through social choice theory, showing that KL-regularization makes the von Neumann winner rule approximately strategy-proof while preserving key axioms like independence of irrelevant alternatives. In contrast, the standard RLHF Borda count rule remains problematic even with regularization, providing theoretical justification for using von Neumann winner objectives in practical alignment.

**Strengths:**

The paper provides an original analysis showing that KL-regularization actually improves the social choice properties of the von Neumann winner rule by making it approximately strategy-proof, while preserving other desirable axioms. This counterintuitive result that regularization can enhance rather than degrade axiomatic properties is a significant theoretical contribution.

The paper bridges abstract social choice theory with concrete alignment algorithms, providing principled justification for preferring von Neumann winner objectives over standard RLHF (Borda count) methods.

The systematic evaluation of multiple social choice axioms (independence of irrelevant alternatives, population consistency, Pareto optimality, strategy-proofness) for both regularized and unregularized versions provides a thorough theoretical characterization that advances understanding of alignment methods.

**Weaknesses:**

The theoretical analysis assumes that individual rater preferences can be cleanly aggregated through pairwise comparisons, but real human preferences may be noisy.

The paper connects to von Neumann winner algorithms like Nash-MD and SPO, it could benefit from discussing how approximation errors, finite sample effects, or optimization challenges in these practical algorithms might affect the theoretical properties.

The paper provides only one synthetic experiment with 3 alternatives and 64 raters using a constructed counterexample from the proof.

**Questions:**

The authors analysis assumes exact computation of preference relations P_H, but in practice these must be estimated from finite preference data. How do estimation errors affect the theoretical guarantees, particularly the strategy-proofness bounds?

---

> ### Author Response · Authors · 2025-11-25
>
> We are glad to see you found our paper providing a significant theoretical contribution and an original analysis with principled justification that shows counterintuitive results that regularization can enhance rather than degrade axiomatic properties. We are truly glad to see your comment that the systematic evaluation of multiple social choice axioms (independence of irrelevant alternatives, population consistency, Pareto optimality, strategy-proofness) in our paper for both regularized and unregularized versions provides a thorough theoretical characterization that advances understanding of alignment methods.
>
> 1. *“The authors analysis assumes exact computation of preference relations P_H, but in practice these must be estimated from finite preference data. How do estimation errors affect the theoretical guarantees, particularly the strategy-proofness bounds?”*
>
> The strategy-proofness bounds for the regularized von Neumann winner are in fact quite robust to estimation errors. The reason for this is Theorem 5.3, which shows that the regularized von Neumann winner is robust to perturbations $H$. Estimation errors can be modeled as such perturbations to $\mathcal{P}\_{\mathcal{H}}$. In this case, one can view estimation errors for pairwise comparisons as being encoded by a perturbation matrix $M$, where the error in the pairwise-comparison of $a\_i$ to $a\_j$ is given by the matrix entry $M\_{i,j}$. Then the resulting perturbation is given by $H(\pi,\pi’) = \pi^{\top}M\pi’$ and $\nabla\_{\pi}H(\pi,\pi’)\rvert\_{\pi = \pi’} = M\pi$. Thus, the approximation error guaranteed by Theorem 5.3 is at most $\frac{1}{4\tau}\lVert M\pi \rVert\_\infty \leq \frac{1}{4\tau}\max\_{i,j}M\_{i,j}$. This means that the overall error due to approximation is bounded in terms of the estimation errors $M\_{i,j}$.

---

### Official Review · Reviewer_bi5D · 2025-10-31

**Soundness:** 4
**Presentation:** 4
**Contribution:** 2
**Rating:** 6
**Confidence:** 3

**Summary:**

The paper studies how KL-regularization to a reference policy alters the social-choice properties ofpreference-based alignment rules. The authors show that the KL-regularized von Neumann winner becomes approximately strategy-proof with individual gains bounded by $O(k/\tau n)$ while it retains independence of irrelevant alternatives (IIA) and population consistency, but only satisfies a “regularized” Pareto notion. In contrast, KL-regularized Borda (i.e., Bradley-Terry/RLHF-style) still violates IIA.

**Strengths:**

- The paper is well-written and a pleasure to read.
- The paper cleanly explains what is and is not preserved under regularization (i.e., IIA and population consistency are preserved but regularized vs. full Pareto). Although the results are not particularly surprising (and have their limitations, see below), the analysis of social-choice-style properties of the von Neumann winner under regularization is well executed.
- I also appreciate that the proofs are presented nicely and easy-to-follow (though they are not very involved).

**Weaknesses:**

- The paper makes unrealistic assumptions about how rater preferences are observed. The whole analysis assumes that we receive every rater’s *full ranking* over all candidates. That is obivously very different from what happens in actual alignment pipelines, where you get noisy pairwise comparisons and the estimation errors / lack of coverage play a crucial role. Even thought the authors motivated their work with the use KL-regularization in practice, this assumption means that there is a large gap between this stylized setting and reality.
-  The main takeaway, that KL regularizing the preference optimization objective improves strategyproofness, is very expected. Since KL regularization limits how far the solution can deviate from the reference policy, it naturally reduces the influence of individual agents. Though, the paper's contribution here lies in the explicit bound on the approximate strategyproofness.
-  The regularized Pareto notion feels a bit unsatisfying. It can violate unanimity if the reference policy assigns low mass to good outcomes, and the paper does not offer a principled way to avoid this issue.

**Questions:**

- Can you say anything about the case where we only observe pairwise preference samples from raters?
- How sensitive are the axioms and bounds to the choice of the reference policy $\mu$? Are there cases where a poorly chosen $\mu$ undermines the guarantees or makes the outcome undesirable? Does the choice matter here at all?
-  The strategyproofness of standard RLHF (i.e., reward maximization/Borda) has been analyzed in recent work [1]. It would be helpful to discuss whether your results connect to or contrast with those findings, especially the known trade offs between social welfare and strategyproofness.

[1] Strategyproof Reinforcement Learning from Human Feedback; Thomas Kleine Buening, Jiarui Gan, Debmalya Mandal, Marta Kwiatkowska, NeurIPS (2025)

---

> ### Author Response · Authors · 2025-11-25
>
> We are truly glad to hear that you found our paper a pleasure to read and well-written clearly explaining what is and is not preserved under regularization where the proofs are presented nicely and easy-to-follow.
>
> 1. *“Can you say anything about the case where we only observe pairwise preference samples from raters?”*
>
> Both the Borda count rule and the von Neumann winner rule can be expressed purely as optimization problems with objectives determined only by expectations over pairwise comparisons $\mathcal{P}\_{\mathcal{H}}(\pi,\pi’) = \mathbb{E}\_{h \sim \mathcal{H}} \mathbb{E}\_{a\sim \pi, b \sim \pi’} \mathcal{P}(a,b)$. See equation (1) for the von Neumann winner, and equation (5) for the Borda count. In other words, both rules we study are optimization problems with a loss that is the expectation over a distribution on pairwise preferences, and hence the loss can be estimated with samples of pairwise preferences from raters. This is the standard setting in essentially all deep learning training, we have a loss we wish to optimize which is given by an expectation over some data distribution, and we then optimize it based on samples from this distribution.
>
> 2. *“How sensitive are the axioms and bounds to the choice of the reference policy $\mu$? Are there cases where a poorly chosen $\mu$  undermines the guarantees or makes the outcome undesirable? Does the choice matter here at all?”*
>
> The choice of the reference policy does matter, though we expect it to be somewhat benign under the most common practical setting where $\mu$ is the pretrained and SFTed model used as the initialization for RLHF. In more detail, if the reference policy $\mu$ puts zero or very low weight on all the most desirable alternatives, clearly the outcome of regularized training will not put much probability mass on these alternatives. However, consider the standard RLHF setting where the policy to be trained is initialized with $\pi\_0 = \mu$ and then RL-trained for $t$ steps to arrive at the final policy $\pi = \pi\_t$. In this case, the regularization just trades off distance from the initialization against performance on the optimization objective. Thus a natural way to think of practical RLHF setups is that we wish to only somewhat perturb the initial policy towards the chosen objective. This is why regularized Pareto optimality seems reasonable in our setting: it simply trades off how far the policy has moved from the initialization against how much each rater prefers that policy.
>
> 3. *“The strategyproofness of standard RLHF (i.e., reward maximization/Borda) has been analyzed in recent work [1]. It would be helpful to discuss whether your results connect to or contrast with those findings, especially the known trade offs between social welfare and strategyproofness.”*
>
> The work [1] studies the setting where each rater’s preferences are assumed to be linear functions, whereas our paper does not make any restriction on how the rater’s preferences arise. The linearity assumption in [1] is useful, in that it allows for a clean algorithm and theoretical analysis utilizing confidence bounds constructed for linear functions. However, there is also the question of how relevant such bounds are to practical RLHF. For example, it is unclear how to construct such bounds for the general deep learning setting, nor is it clear what deep-learning algorithm one should attempt to use if one wants to obtain strategy-proofness for standard RLHF in practice. The goal of our paper is to analyze practical RLHF using the lens of social choice theory, and thus we focused on how the axioms of social choice could be directly applied to the different practical RLHF algorithms currently in use. We will add a citation and discussion of the relevance of [1] to the related work section.

---

### Official Review · Reviewer_Ysbu · 2025-10-31

**Soundness:** 2
**Presentation:** 3
**Contribution:** 2
**Rating:** 4
**Confidence:** 3

**Summary:**

The paper shows that Nash equilibrium-based preference optimization, when coupled with KL regularization, has much stronger strategy-proofness against misreports of preference, and validates that with numerical experiments. It then shows that classical methods for preference optimization continues to suffer from violation of independence of irrevelant alternatives after KL regularization.

**Strengths:**

- Soundness: I followed the derivations/proofs in the body and found no error. I did not check the appendix.
- Clarity: The motivation and approach is clear. The structure of the paper is clear and the writing is accessible.
- Significance: There have been incidents of data poisoning of language models, so strategy-proofness of alignment is a timely topic (although the authors included limited experimental validation). Introducing regularization into the picture makes the setup significantly more realistic, and may be a substitute for complicated ways of modeling structures (e.g. Rademacher complexity) of the hypothesis class in the social choice theory of alignment.
- Originality: I am not aware of any prior work with significant overlap.

**Weaknesses:**

- The claim in the abstract (and elsewhere) that "the standard RLHF objective [...] offers no such improvement" is potentially misleading. The paper showed that regularization does not restore IIA for Borda count, but does not rule out, e.g., approximate IIA, or approximate strategy-proofness. When the core message of the paper is comparing vNw against BC and claiming that regularization strengthens the former's position in an asymmetrical way, it would make sense to compare the impact of regularization on vNw vs BC on comparable dimensions.
- The empirical experiment is a minimal one, essentially a numerical replication of the main theorem statements. It will be helpful to see the extent to which the theory aligns empirically with language modeling experiments, by comparing the impact of distorted preference data on MD/SPO-trained vs RLHF/DPO-trained language models.

**Questions:**

Regarding Section 7 (Experiments):
- What is the reason for using Nash MD for regularized training, and SPO for unregularized training? There seems to be no obvious reason why Nash MD can be regularized while SPO cannot.

Regarding practical relevance:
- One traditional justification for approximate strategy-proofness is that the cognitive cost for computing how to misreport preferences outweighs the gains from misreporting [1]. In the case of preference alignment for language models, is this true? What are the simplest examples of good strategies for misreporting preference, when it comes to language model training?


[1]  Approximately Strategy-Proof Voting

---

> ### Author Response · Authors · 2025-11-25
>
> We are glad to see you noting that our paper is on a timely topic with a clear and accessible structure with sound proofs and derivations and original contributions. We appreciate the time you have invested in reviewing our paper.
>
> 1. *“The claim in the abstract (and elsewhere) that "the standard RLHF objective [...] offers no such improvement" is potentially misleading. The paper showed that regularization does not restore IIA for Borda count, but does not rule out, e.g., approximate IIA, or approximate strategy-proofness. When the core message of the paper is comparing vNw against BC and claiming that regularization strengthens the former's position in an asymmetrical way, it would make sense to compare the impact of regularization on vNw vs BC on comparable dimensions.”*
>
> Thank you for pointing this out! In fact, the proof of Proposition 6.1 immediately applies to approximate IIA. In particular, consider the following definition of approximate IIA. In the setting of Definition 4.2 (IIA) let $\mathcal{R} = F(\mathcal{B},\mathcal{H})\rvert\_{\mathcal{A}}$ and $\mathcal{R}’ = F(\mathcal{B},\mathcal{H}’)\rvert\_{\mathcal{A}}$. The rule $F$ satisfies $\epsilon$-approximate IIA if for every $\pi,\pi’$ such that $\mathcal{R}(\pi,\pi’) \geq 0$, then there exist policies $\hat{\pi},\hat{\pi}’$ with $D\_{\text{TV}}(\pi,\hat{\pi}) < \epsilon$ and $D\_{\text{TV}}(\pi’,\hat{\pi}’) < \epsilon$ such that $\mathcal{R}’(\hat{\pi},\hat{\pi}’) \geq 0$.
>
> Intuitively, this definition says that the relative ranking over $\mathcal{A}$ produced by $F$ is not sensitive to changes in irrelevant alternatives, so long as we are allowed to perturb the policies in question by up to $\epsilon$ in total variation distance. That is, if $\pi$ is preferred to $\pi’$ by $\mathcal{R}$, then there are two nearby policies $\hat{\pi},\hat{\pi}’$ such that  $\hat{\pi}$ is preferred to $\hat{\pi}’$ by $\mathcal{R}’$. The proof of Proposition 6.1 in fact shows that regularized Borda count does not satisfy $\epsilon$-approximate IIA for any $\epsilon < 1/2$. We have updated the paper to reflect this.
>
> 2. *“What is the reason for using Nash MD for regularized training, and SPO for unregularized training? There seems to be no obvious reason why Nash MD can be regularized while SPO cannot.”*
>
> The reason we used Nash MD for regularized training and SPO for unregularized is that this is the way both algorithms were initially proposed, and we wanted to make the connection as clear as possible to the original practical RLHF papers that proposed these algorithms. If we had instead designed a regularized variant of SPO, this perhaps would have seemed like we were modifying existing algorithms to fit into our framework, whereas algorithms for both regularized and unregularized versions of the vNw rule had been proposed previously. We can clarify this point in the paper, as one could of course train a regularized version of SPO.
>
> 3. *“One traditional justification for approximate strategy-proofness is that the cognitive cost for computing how to misreport preferences outweighs the gains from misreporting [1]. In the case of preference alignment for language models, is this true? What are the simplest examples of good strategies for misreporting preference, when it comes to language model training?”*
>
> A similar argument that the costs for computing how to misreport will outweigh the gains should also hold here. The context is quite similar, individual raters make pairwise comparisons of language model responses, which are aggregated to align the final trained language model policy. In order to misreport effectively, a rater must know the aggregate preferences of the other raters, and then compute a deviation which changes the outcome. This is much as in the original model of computational social choice, where to misreport a voter must compute a deviation from their true preferences in order to change the outcome, while taking the behavior of all other voters into account.
>
> For a simple example, suppose that a language model is being trained to explain algorithmic concepts. Such concepts can be explained with mathematical formulas, with pseudocode, and with diagrams, and each rater may have preferences for one type of explanation over another. In this case, an individual rater who prefers diagrams over pseudocode, and pseudocode over formulas may benefit by incorrectly reporting that they prefer formulas over pseudocode. In fact, Theorem 5.5 shows that, if the other raters are in aggregate “cyclic” in their preferences over the three types of explanation, then under the unregularized von Neumann winner rule, it makes sense for this individual rater to deviate, as it would greatly increase the proportion of diagrams in the LLM policy’s response.

---

### Meta-Review · Area_Chair_myUS · 2026-01-05

**Summary:**

The reviews consistently acknowledge that this is a well-written, theoretically careful paper that cleanly characterizes the effect of KL regularization on social-choice axioms, with sound proofs and a clear conceptual contribution. However, there is also a shared concern that the practical implications remain narrow and largely illustrative, rather than impactful for real alignment pipelines, across the reviewers. The experimental section is exceptionally minimal—amounting to small synthetic numerical replications of the main theorems (e.g., three alternatives, stylized preference matrices)—and does not meaningfully test the theory in realistic language-model alignment settings, a gap explicitly noted by multiple reviewers (Ysbu, fHB9, bi5D). While the authors convincingly argue that their results are robust to estimation error and that pairwise samples suffice in principle, this does not substitute for empirical evidence showing how the proposed guarantees manifest under noisy, sparse, prompt-conditioned preference data or modern RLHF training dynamics. As a result, the paper’s main takeaway—that KL regularization improves strategy-proofness of von Neumann winner–style objectives—is theoretically expected and carefully formalized, but does not translate into concrete guidance or demonstrable advantages for practitioners, especially relative to standard RLHF methods. Given the limited experimental validation and the large gap between the stylized setting and real-world alignment pipelines, the overall contribution feels too narrow for acceptance.

**Reviewer Concerns:**

See above.

**Reviewer Scores:**

See above.

---

### Decision · Program_Chairs · 2026-01-26

Reject